# How Primary School Curriculums in 11 Countries around the World Deliver Food Education and Address Food Literacy: A Policy Analysis

**DOI:** 10.3390/ijerph19042019

**Published:** 2022-02-11

**Authors:** Kim Smith, Rebecca Wells, Corinna Hawkes

**Affiliations:** Centre for Food Policy, City, University of London, London EC1V 0HB, UK; rebecca.wells.1@city.ac.uk (R.W.); corinna.hawkes@city.ac.uk (C.H.)

**Keywords:** food literacy, food education, primary school, food education policy, food systems

## Abstract

(1) Background: As one of the biggest drivers of health and climate change, the food system has unrealised potential to influence consumption toward affordable, healthy, sustainable diets. A range of policy levers, including mandating food education, are needed. Schools are considered the best place for food education and childhood is a crucial period when eating habits that persist into adulthood are formed. Food education as part of the curriculum is crucial in generating population shifts in food systems improvements. The purpose of this policy analysis was to analyse mandatory curriculums in different countries to explore the ways in which primary school food education addresses food literacy. (2) Methods: This study analyses how food education within primary school education policy, in 11 countries, addresses Food Literacy (FL). It is the first study of this kind. A case study methodology was employed, and curriculum policy content analysis was conducted using a Food Literacy framework. (3) Results: Each country has a curriculum dedicated to food education, supported by food education in non-food curriculums. There is no standardized approach to primary school food education policy, no consensus in primary food education nomenclature or what curriculums constitute. Curriculums focus on cooking and health topics, but significantly less on social-cultural, equity, and sustainability issues. (4) Conclusion: How primary curriculums around the world deliver food education policy to address FL varies enormously. All 11 countries have dedicated food curriculums, supported by non-food curriculums, but there is no consensus as to what food education is called or constitutes. Countries rarely deal with FL comprehensively. The most comprehensive are single, detailed food curriculums, complemented by non-food curriculums where food knowledge and skills progress clearly and are the intended learning outcome.

## 1. Introduction

Incontrovertible evidence places the food system (FS) as the single biggest driver of global challenges including malnutrition in all its forms, environmental degradation, and 30% of global greenhouse emissions [1,2]. However, this same FS has the potential to be transformed and influence consumption of affordable, healthy, sustainable diets for all [3]. Transformation of such a dynamic, interconnected system requires holistic consideration for effective solutions through a diverse combination of hard and soft policy levers including taxes, legislation, labelling, and education [4].

Education, recommended as a precursor to overcome inevitable resistance to stronger policy measures [3], can ensure society plays its part in FS transformation. High-level governance considers teaching people about the links between food, health, and the environment and how to consume a sustainable diet as enabling action for change towards Sustainable Development Goals [5]. Armed with understanding, people can leverage their capacity for improving the FS, and its impacts on human and planetary health.

Schools are widely considered the best environment for food and health education [6,7]. Childhood is a crucial period when food and lifestyle habits that may persist into adulthood are formed [8], and when ecological awareness is developed [9]. Evidence suggests food education from primary school to adolescence has the potential for positive health, cooking, and sustainability behaviours to track into adulthood [10,11,12]. Schools present multiple opportunities to learn about food, but policy and recommendations often centre on school meals [3,13]. Whilst mealtimes have a place in pedagogy [14], according to the WHO [8,15] a comprehensive food education should be implemented through curriculum policy, but this has largely not been implemented [16,17].

What is meant by “food education” varies widely across research and policy domains, as shown by the range of terms collated in Table 1. This highlights the lack of nomenclature consensus and shows how each term originates from a different perspective, subsequently determining topics included. This results in a lack of consensus over what food education constitutes. For example, “Home Economics” (HE) focuses on the domestic food setting, whilst “Food Technology” is science and design-based, with an industrial focus replicating food manufacturing. “Nutrition Education” focuses on health knowledge, whilst “Cooking Skills” focuses on practical planning and techniques. A more generic term like “Food Education”, devoid of perspective, encompasses anything related to food, whilst “Food Literacy” (FL) has a broad perspective, including sustainability and sociocultural topics, combining food and health knowledge in conjunction with practical cooking skills.

Whatever food education is called, analysis of what takes place in the curriculum is limited and devoid of government review. The most significant research, albeit grey literature, analysed England’s cooking and nutrition curriculum policy [16]. It discovered food education implemented through the curriculum to be largely inadequate, with variation across schools in food education content, quality, and quantity. The curriculum and implementation lacked a comprehensive approach to food, unlike other subject curriculums such as mathematics, which develop subject progression of learning. It found primary food education implementation to be particularly poor, with less than half of primary school children experiencing cooking lessons more than twice a year [16]. Notably, the research lacked definition of food education terms and focused on cooking and healthy eating. England’s education inspectorate Ofsted [17] similarly found in their childhood obesity review that “For something that is explicitly set out in… curriculum, the proportion of children being taught to cook is very low”.

But this is not the case in every country. The lead author’s experience at a UK food education charity, has revealed alternative approaches, such as Finland’s sensory food education [25], whilst international secondary food education research also indicates other food education curriculum approaches exist [26].

This paper will firstly summarize food education policy, food education, and food literacy literature, then explain the methodology used to examine primary curriculums in 11 countries. Findings from each country are presented under themes of food education curriculums, non-food curriculums, how countries address food literacy (FL) and the impact of policy on FL.

### 1.1. Food Education Policy Literature

Literature examining food education policy is minimal, and what exists is broad in nature, covering policy in primary or secondary education, spanning all levels of governance from EU food policy mapping [13] to papers concerning national food education policy [27]. They either evaluate food education policy implementation [28,29]; focus on national food education policy analysis [26,30,31]; or compare policy across multiple countries [13,32,33]. There is a lack of research concerning primary food education policy, content analysis of current food education curriculums, and international examples of food education policy approaches.

### 1.2. Food Education Literature

Contrastingly, there is a very substantial body of research about food education interventions spanning pre-school to graduate school (Appendix A, Table A1). It covers an extensive range of conventional food topics such as gardening, growing fruit and vegetables, cooking and health whilst newer approaches such as sustainability and sensory food education indicate the field‘s continued evolution. There are substantially fewer primary school interventions, but they are equally diverse in range. Papers predominantly focus on nutrition education, cooking or healthy eating [34,35,36]. They frequently measure health improvements or nutrition knowledge outcomes [37]. No studies analysing primary school food education curriculum content, quality, or comprehensiveness were found.

Increasingly, studies do combine two topics like cooking and healthy eating, or growing fruit and vegetables and cooking [38,39]. A systematic review examining garden-based nutrition programmes for their impact on children’s nutritional outcomes found, despite scant evidence, that such programmes have potential to influence children’s fruit and vegetable consumption [40]. Meanwhile, research into a 10-week cooking and nutrition session with 271 US elementary school children, discovered chef-led sessions improved nutrition knowledge and cooking efficacy [41].

A few multi-component food education interventions include combinations of cooking, growing fruit and vegetables, nutrition, school meals, parent engagement sessions, sensory education, physical education, and sustainability. The most comprehensive of these, in all but one study, were randomized controlled trials designed to measure child health outcomes from a specific programme, outside of the primary academic curriculum, for a fixed period [36,42,43,44].

Notably, food education research usually focuses on interventions that take place outside of the mandatory curriculum, are short or fixed term and lack progression of learning throughout the primary years, unlike other subjects such as mathematics. A few studies do explore food education taught within mandatory curriculums [45,46,47]. One study [48] used food to teach mathematics and science curriculums in hands-on sessions with 4th grade students finding the intervention more effective than the control in increasing food and multidisciplinary science knowledge. However, food knowledge advancements are only a supplementary outcome.

Another evaluation of a civil society, whole school approach including gardening, cooking, and nutrition taught by invited specialists, not teachers, found such programmes may improve diet quality, fruit and vegetable consumption and raise awareness of sustainability [44]. Yet, despite several studies researching interventions combining more than one food topic, no studies examining comprehensive food education covering all the topics to equip children with the knowledge and skills to navigate the FS, were identified.

### 1.3. Food Literacy

A concept beginning to address the lack of comprehensive food education is the emerging and evolving concept of food literacy (FL). Studies to date develop definitions [24,49,50,51], create measurement tools or measure population FL levels [52], and form the basis of school interventions [53,54,55]. Research is beginning to show that an increase in FL may increase nutrition-related diet quality in children [56,57].

Since 1990, research has increasingly endeavoured to encapsulate the full spectrum of activities associated with the apparently simple act of eating, recently expanding to include FS [58]. Cullen et al.,’s [24] (p. 143) definition reflecting FS thinking defines FL as:
*the ability of an individual to understand food in a way that they develop a positive relationship with it, including food skills and practices across the lifespan in order to navigate, engage, and participate within a complex food system. It‘s the ability to make decisions to support the achievement of personal health and a sustainable food system considering environmental, social, economic, cultural, and political components.*

In application, FL studies examine the interaction between FL and home economics [59,60,61,62] or measure FL. The latter, small but growing body of literature measures FL levels to determine health outcomes and diet quality in adults [63,64,65,66] adolescents [56,57,67,68,69,70] and children [71,72,73]. The tools to measure FL in populations include questionnaires [63,74]; a model for incorporating FL in schools [75]; and a competency-based framework developed for application in schools and communities [76]. The only application of FL in policy is found in the grey literature analysis of the impact of Canadian policies on FL, concluding with policy implications [31].

Food literacy is grounded in the understanding that learning to eat well is complex. FL has become increasingly comprehensive and considers knowledge, skills and relationships needed to navigate the FS across their lifespan, applying FS thinking to integrate social-cultural, environmental, political, and economic elements. To date application of FL focuses on individual measurement or interventions, but its comprehensive approach makes it a fitting framework for evaluating food education policy. Examining the way in which curriculums currently address FL could highlight FL gaps, improve FL outcomes, and potentially contribute to personal and planetary health improvements.

### 1.4. Study Purpose

As outlined above, food education as part of the curriculum is crucial in generating population shifts in food systems improvements [16,17]. Understanding of international primary school food education policy is lacking, especially examples of comprehensive curriculums that educate children how to navigate the FS. Studies examining food education policy are rare, particularly in primary schools, or mandatory curriculums. FL research is evolving but is yet to be applied in food education policy.

For these reasons, the aim of this study was to analyse mandatory curriculums in different countries to explore the way in which food education within primary school curriculums addresses FL. Using FL as a proxy for a comprehensive food education, it investigates whether current food education policy is comprehensive enough, to help children navigate the food system, for life.

Specifically, the research aimed to:select and develop a framework to analyse how well each country addresses FL.analyse primary education curriculums that include food education from different countries, using the framework.identify insights from primary education curriculums in other countries to improve food education policy.

## 2. Materials and Methods

This study formed part of a master’s dissertation and was a holistic, multiple-case study of primary school education curriculums from different countries, and originally employed a case study methodology. A national primary school education curriculum implemented in each country between April and December 2020 was classed as a single case.

The case study samples were selected through internet searches for mandatory primary school curriculums for children aged 4–11 years, against inclusion and exclusion criteria (Table 2) determined during the scoping exercise. Data were collected through content analysis of secondary data in the form of policy documents. The content of each education policy document was analysed against a FL framework to establish the amount and forms of food education in each country’s curriculum policies to evaluate the ways in which a country addresses FL. Here we present a summary of how the 11 countries include food education in curriculums, how they address FL, and the impact of policy approach on how countries address FL.

### 2.1. Policy Content Analysis

An overview of the policy content analysis stages is detailed in Figure 1 and Figure 2.

A brief scoping exercise verified whether enough quality, publicly available data from different countries was available in English to analyse. Initially, a broad internet search of countries around the world was undertaken to find as many different primary curriculums as possible that contained food education (step 1). Internet searches were conducted to find publicly available, national, or federal government education policies that specifically referred to any form of food education.

Search terms from Table 1 in combination with the terms “national”, “school”, “curriculum”, “policy”, and “education” were employed, alongside insight from literature, to uncover potential countries where food education might be primary education policy. Several government websites and curriculums were available in English, such as Norway’s government website and Sweden’s curriculum. However, if websites and curriculums were not available in English, several approaches were employed. First Google Translate was used directly on the website to enable the correct documents to be found. In some instances, if after extensive searching the relevant curriculums were not discovered, the location was removed from the search. If after finding the relevant documents, they required translation, for example in Slovenia, the documents (often PDF’s) were copied and pasted into a Word document and translated using Google Translate. On some occasions the format of the document prevented them being copied and translated, so were subsequently excluded from searches, such as in Bulgaria (step 2).

This resulted in a convenience sample of 26 countries (including the USA, Cyprus, Greece, Austria, Germany, Bulgaria, Japan, and Singapore) and 6 federal states (including Quebec, Victoria, New York) where food education was included in primary education curriculums. A spreadsheet was used to organize potential countries and curriculums (step 3).

Curriculums are defined as “the official, mandatory statement of what is to be taught to students” [77] (p. 225). These documents, direct from national and federal government websites were searched for and subsequently analysed. The lead author’s experience working in UK schools indicated that food education might take place in non-food subjects like science, therefore all primary subject curriculums were included in analysis.

To keep research within scope, an inclusion criteria was developed using the 26 countries and 6 states gathered in the scoping exercise. Curriculums were analysed thoroughly for common themes, such as whether a curriculum was national, mandatory, or current policy (step 4). Consequently, 11 places met the inclusion criteria including Australia, Czech Republic, Denmark, England, Iceland, Ireland, Japan, Norway, Scotland, Slovenia, and Sweden. Wales, whilst it met the inclusion criteria, was omitted as it had a new curriculum under development (step 5).

Notably, searches resulted in an absence of low- or middle-income countries, despite broadening search terms to specific countries such as India and Chile, known for food education policy. Similarly, countries such as Italy and France with strong food cultures were expected to be included. Analysis identified the criteria for mandatory food education in curriculums, and curriculums specifically for primary (Table 2) determined why such countries were excluded from analysis.

### 2.2. Framework Development and Scoring

A framework was developed to facilitate analysis, by evaluating frameworks across FL research for their appropriateness to address the research aims (step 6). Albeit intended for use with young adults, the framework (Figure 2) [76] was the most comprehensive and reflected the most developed form of FL. Evaluation against the FS model (Appendix B, Figure A1) also confirmed the framework as the closest representation of an FS approach.

Framework development continued into data collection as important findings from one case study can prompt the researcher to reconsider the design [78]. Initial curriculum analysis revealed competencies not included in the framework. These, listed below, were added to the framework (Figure 2), employing similar language from the original framework. Previous cases were revisited accordingly.

Understanding different life stages (e.g., food needs of babies, children, and elderly).Understanding current healthy eating guidelines and can apply them to their own dietUnderstanding how to grow food in a garden.Understanding the role of the senses in eating.Understanding the FS.Understanding the role of politics and economics in the FS.

Both the original FL competencies (Figure 2) as well as the additional competencies above were converted into a FL framework for scoring, using Excel, whilst maintaining wording, themes (Confidence and Empowerment with Food, Joy and Meaning through Food and Equity and Sustainability for Food Systems), subgroups and competency structure from the original framework (Appendix C, Table A2).

A trial analysis (step 7) using England‘s curriculum was conducted to test study design, framework, and outcome quality. Consequently, a scoring guide (Table 3) was developed to reduce subjective bias, make scoring more transparent, illustrate nuance in how curriculums addressed FL and accommodate disparity in language between broad policy terms and specific competency wording (step 8). Trial analysis also highlighted how some curriculum topics with potential to address FL, such as sustainability, must be explicitly connected to food in the document, to be scored.

### 2.3. Curriculum Content Analysis

Curriculum documents from 11 countries were analysed systematically (Figure 3) (step 9).

Prior to analysis, spreadsheets for each country were created, with columns to record FL framework scoring (step A). Each FL framework theme (Confidence and Empowerment with Food, Joy and Meaning through Food, and Equity and Sustainability for Food Systems), was allocated a colour to highlight associated words in curriculum documents, facilitate scoring, and ensure all relevant food education was captured (step B).

Analysing one country at a time, each subject in that county’s primary curriculum was analysed for evidence of food education. When any food education was identified it was coded with the appropriate theme colour (step C). Subject curriculums without any food education were omitted from further analysis.

Subject curriculums with food education identified were then analysed using the FL Framework (step D). Each mention of food education was scored using the scoring guide. This was repeated for each curriculum (step E). Once all relevant curriculums in a country were analysed, this was repeated for other countries (step F).

### 2.4. Data Analysis

Each country was analysed as a single case study to establish the ways they address FL (step 10). Curriculum findings were interpreted separately, then how the curriculums work together to address FL. Analysis of each curriculum was based on the FL framework structure (Appendix C). Case studies were analysed independently, not compared.

Graphs present data collected through scoring each curriculum against the FL framework. FL competencies are grouped by subgroup rather than presenting all 62 FL competencies. Subgroups are presented under the original framework themes (Confidence and Empowerment with Food, Joy and Meaning through Food, and Equity and Sustainability for Food Systems) shown across the body of each graph. The same colours used in coding curriculums were used in the graphs. Health curriculums are red, practical approach curriculums are blue, science curriculums are yellow, and social sciences are green.

The scores were calculated by first establishing the total possible score for each FL subgroup. For example, the Nutrition Knowledge subgroup had 9 FL competencies and a maximum score of 3 available from the scoring criteria, which gave a total possible score of 27 for Nutrition Knowledge. Each curriculum score was then converted into a percentage to remove variation in how many competencies each subgroup contained. For example, Australia’s D&T curriculum Nutrition Knowledge score was 9 out of 27, represented on the chart as 33%.

## 3. Results

### 3.1. General Features of Food Education in 11 Countries

Each country had between 2 and 4 different curriculums that included food education (policies detailed in Appendix D). These were food and non-food curriculums (Table 4). Norway is unique with a generalist food curriculum where food is the central focus topic. In all other countries food is only one part of the curriculum which has a different central learning focus. For example, a subject such as D&T uses food as a material in the design process; broad health curriculums cover mental, sexual, and emotional health alongside healthy eating and vocational curriculums, as in the Czech Republic’s Humans and the World of Work curriculum, which considers food alongside topics such as construction.

### 3.2. Food Curriculums

All countries had a dedicated food curriculum, where food was a central or dominant topic. Curriculums were either practical in nature—Home Economics (HE), Design and Technology (D&T)—or health oriented—Health and Physical Education (HPE), Health and Wellbeing (HWB). Curriculums are largely called different things, inherently determining curriculum content as indicated in FL scoring variation. Most countries have a single food-focused curriculum. Slovenia, Japan, Czech Republic, Iceland, Norway, and Sweden all have practical, focused food curriculums, whilst Denmark and Ireland have health-focused food curriculums. However, England, Australia, and Scotland employ both types of food curriculum.

As the graphs for each country show (Figure 4), all countries predominantly focus on FL Theme 1 (Confidence and Empowerment with Food). Within Theme 1, England, Iceland, Denmark, and Ireland focus on the Nutrition Knowledge subgroup and multiple curriculums cover the same healthy eating competencies. The Food Preparation Skills subgroup was comprehensively addressed in Slovenia, Iceland, Sweden, and Scotland, whilst Norway’s Curriculum in Food and Health (CFH) addressed all bar one Food Preparation Skills competency comprehensively. Norway uniquely addresses digital skills to “strengthen the practical cooking skills” [79]. All countries addressed the Hygiene competency, but only 4 addressed Budgeting (Iceland, Japan, England, Slovenia). Understanding the Role of Media on Food Choices (under both Themes 1 and 3) is an area that England, Iceland, Japan, and Czech Republic do not address.

FL Theme 2 (Joy and Meaning through Food) was addressed to a lesser extent by most countries, whilst the Czech Republic did not address any competency in this theme at all. Each country addressed a different selection of competencies. Enjoying Cultural Foods and Having a Positive Relationship with Food are the subgroups most frequently addressed, the latter often through reference to wellbeing, to varying degrees of comprehensiveness. Scotland and Norway most comprehensively addressed Theme 2, although coverage was still not complete. Norway uniquely addresses 12 of 15 competencies through emphasizing the pleasure of cooking and eating with others, social and gender equality, cultural diversity, and the value of this for self-esteem and community.

Under Theme 3 (Equity and Sustainability for Food Systems) every country except England and Japan addressed at least one of twelve competencies within this theme, although rarely comprehensively. Lobby and Media Influence was the subgroup most often addressed whilst Norway was the only country to address Understanding the Food System subgroup. Australia, Norway, and Scotland most comprehensively addressed this theme. Scotland’s HWB explicitly connects the FS with wider impacts: “When preparing and cooking a variety of foods, I am becoming aware of the journeys which foods make from source to consumer, their seasonality, their local availability, and their sustainability” [80] (p. 12). Overall, only Norway, Scotland, and Sweden explicitly connect FS impacts with planetary and public health. As Norway states, “Food and health must contribute to the students developing critical thinking, ethical awareness, and a sense of responsibility so that they are able to choose food that is both health-promoting and sustainable” [79].

### 3.3. Non-Food Curriculums

Examining every subject curriculum for food education confirmed the lead author’s experience of food education in non-food subjects. The findings shown in Figure 4 show that food education is included in non-food curriculums. It was discovered within science in all countries (Table 5). Science in every country includes plant knowledge, addressing the Understanding How To Grow Food competency, and everywhere except Japan and Australia includes human health. Half of the countries address Hygiene such as food safety within science. Denmark uniquely includes practical food work in science where children “put together health-promoting meals on the basis of relevant dietary recommendations” [81].

Other non-food curriculums also address food literacy (Table 4). Japan uniquely uses PE to teach healthy eating, whilst six countries address FL using social science. Scotland, Japan, and Iceland use a broad social science curriculum to address unique FL competencies such as local food and agriculture. Scotland includes identifying “forms of agriculture in Scotland and foods associated with these, for example, arable, dairy or pastoral” [82]. Sweden and Ireland employ a specialist Geography curriculum, whilst the Czech Republic has a Cross-Curricular Subjects curriculum each addressing FL through agriculture and food’s environmental impact. Ireland highlights the impact of the national food and farming industry on people and the environment, through learning about food waste or exploring the role of trade on food commodities. Notably, social science curriculums frequently have the potential to include multiple FL topics such as ethics, economics, politics, media, social justice, and the environment, but FS links are rarely made, meaning it cannot be scored for addressing FL.

### 3.4. How Curriculums Address FL Together

When curriculums are considered together in how they address food literacy overall, each country has a unique approach addressing FL to varying degrees of comprehensiveness. Norway, Scotland, Slovenia, and Australia each addressed FL extensively, covering all three FL themes, and addressing over 30 out of 62 FL competencies.

Iceland has comprehensive coverage of Theme 1 (Confidence and Empowerment with Food), but less comprehensively addresses other themes. Whilst Denmark, England, Japan, the Czech Republic, Ireland, and Sweden also focus on Theme 1, they have patchy coverage across Themes 2 (Joy and Meaning Through Food) and 3 (Equity and Sustainability for Food Systems).

Several countries’ curriculums over-index on the Nutrition Knowledge subgroup at the expense of addressing other FL competencies. This is particularly the case in Iceland, Denmark, Japan, Ireland, and England. Contrastingly, Norway does not focus heavily on nutrition and omits nutrition, food groups, portioning, and different dietary needs competencies.

### 3.5. Impact of Policy Detail and Language on FL

Through examining all curriculums in 11 countries, it is evident that the policy approach, progression of learning and level of policy detail influences how comprehensively curriculums address FL.

Many countries employ cross-curriculum themes such as literacy, which permeate every subject. However, Australia and Norway uniquely use this policy approach to address sustainability, health, and cultural topics. Australia‘s cross curricular priorities for Inter-cultural Understanding and Sustainability [83] permeate all curriculums, bringing consistent emphasis on sustainability and native cultural food practices into each subject. This significantly improves how socio-cultural and sustainability themes are addressed. Likewise, in Norway each curriculum is underpinned by interdisciplinary themes of Public Health and Life Skills, Digital Skills and Sustainable Development, which helps Curriculum for Food and Health extensively and broadly addresses FL.

Clear progression of food education knowledge and skills from the beginning to the end of the primary phase, influences how FL is addressed. In Australia and Scotland there is clear progression across year groups with extensive detail, advancing in complexity across each year of primary. Consequently, more FL competencies are addressed. Scotland’s HWB states Early Level starts off with “simple food preparation techniques, for example, peeling, slicing, mixing, spreading” progressing by Second Level to “weighing and measuring, kneading, chopping, baking, grilling” [84] (p. 4). Contrasting this in England, the D&T curriculum gives scant, broad and very similar curriculum content for the two phases of primary (KS1 and KS2) with minimal explicit advancement in food knowledge and skills.

Similarly, the level of structure and detail within curriculums affects how countries address FL. Australia, Scotland, and Slovenia had detailed curriculums which broke down large topics into specific smaller items to be covered in each year of primary school. Scotland‘s HWB includes Food and Health as one of three topics in the curriculum, alongside PE and mental, emotional, social, and physical wellbeing. However, it goes on to further break down Food and Health into “The food experience; Developing healthy choices; Nutritional needs; Keeping safe and hygienic; The journey of food; and Technologies” which each detail specific and detailed learning outcomes for each year of primary [80].

Contrasting this, England, Japan, Sweden, and the Czech Republic‘s basic curriculums lack explicit detailed explanations. England includes Cooking and Nutrition as part of the D&T curriculum meaning food education is taught through a design lens, with food knowledge a supplementary learning outcome and with few details. Despite acknowledgement of many FL competencies, vague statements such as “understand and apply the principles of nutrition and learn how to cook” [85] (p. 1) lack elaboration and therefore do not address many FL competencies.

## 4. Discussion

The purpose of this research was to analyse the way in which mandatory primary curriculums around the world employ food education to address FL, to identify if food education curriculums can be comprehensive enough to help children learn to navigate the FS. These findings address two major gaps; it is the first study of primary school food education policy around the world and is the first application of FL to evaluate different countries’ primary curriculums. This discussion separately considers how different countries address food education and FL, then examines the influence of policy detail and language on FL.

### 4.1. Food Education Findings

#### 4.1.1. All Countries Have a Dedicated Food Education Curriculum Where Food Is a Core Topic, but There Is No Standardised Approach to Topics Included in Food Education

Given the widely held view that children rarely learn about food at school [86] it was unexpected to discover all countries under investigation had a mandatory, dedicated food curriculum, (Table 4). This confirms food education is included within mandatory, primary academic curriculums and addresses FL. This differs from existing literature that examines food education as an “add-on” to the curriculum and begins to address the lack of research about food education within curriculums.

Each studied country has a unique approach to food curriculums. All employ a combination of food and non-food curriculums, but this is where similarities end. Country curriculums each address different collections of competencies, take either a health or practical perspective. Some have a minimal food education in the curriculum, whilst others have detailed, comprehensive and broad curriculums. Contrary to insight from the literature, collectively these findings show food education is addressed within academic curriculums, but a standardised approach to topics is lacking, unlike in other curriculums, such as mathematics.

#### 4.1.2. There Is No Consensus in Primary Food Education Nomenclature and What Curriculums Constitute

Different countries call their food curriculums by different names. Iceland has HE, Australia has D&T, whilst Scotland has HWB (Table 4). This validates the lack of consensus found in the literature about food education terms. However, it differs from secondary food education findings where use of the term “Home Economics” globally, contributes to nomenclature consistency [33].

Analysis also revealed a pattern in nomenclature with food curriculums either health or practically orientated. Health curriculums include HPE and Relationships and Sex Education and Health Education (RSE&HE) (Ireland, Denmark), whereas practical curriculums include HE and D&T (Japan, Norway, Sweden, Slovenia, Iceland, Czech Republic).

This is more than just a name. Such variation in nomenclature consequently determines the curriculum content, which in turn impacts the types of food education taught and how comprehensive it is. For instance, Iceland‘s HE concentrates on food preparation skills, consumer habits, and economic literacy, whereas Ireland‘s SPHE focuses on healthy lifestyles, nutrition, and implementing dietary guidelines, both with very different food learning outcomes. Such wide curriculum content variation is unlike other subject curriculums. In Science, for example, there is general international consistency in what constitutes a basic scientific education. Yet in primary food education, there is no such global consensus, consequently linked to a lack of international food education subject standardization.

Three countries take a different approach. Australia, England, and Scotland use two curriculums, employing both health and practical curriculums, although resulting in different FL outcomes. Australia‘s D&T is relatively complementary to the HPE curriculum, addressing FL more comprehensively. In Scotland, HWB is a bigger and more comprehensive food curriculum, supported by a narrower range of competencies in Technologies, although it still results in an equally comprehensive FL approach. England however over-indexes on health in both curriculums, leaving fewer FL competencies addressed and inadequate FL overall.

By establishing what primary food education curriculums are called (Table 4) this research has uncovered the impact on curriculum content, an issue which until now, was unidentified. Using the FL framework highlights a distinct lack of consensus over what food education policy is called and includes. This also presents a solution. Aggregating all food subtopics within a food literacy curriculum could reduce nomenclature variation, broaden food education curriculum scope, consequently creating a comprehensive, primary food education that teaches children to navigate the FS.

#### 4.1.3. Non-Food Curriculums Include Food Education

Non-food curriculums such as science, social science, geography, and PE support dedicated food curriculums in all countries (Table 4). Largely complementary, they contribute unique FL topics like agriculture or nutrition science. Both the findings and literature illustrate how food’s broad nature lends itself to food education across different subjects, an emerging approach being taken for example, where science is taught using food [48]. There is still abundant scope for non-food curriculums to contribute to food education, as many FL competencies were not scored due to their lack of food links.

#### 4.1.4. Food Curriculums Focus on Knowledge and Lack Practical Skills Development

FL recognises that both skills and knowledge are needed for people to eat well [87]. Findings, however, indicate many countries prioritise knowledge-based learning, leaving practical skills such as budgeting, food preparation, digital skills, and shopping inconsistently unaddressed. The Hygiene subgroup, addressed in all countries, is the exception. Whilst all countries, except Ireland, included some form of Food Preparation Skills, competencies under Making Healthy, Economical Food Choices, and Budgeting subgroups were not comprehensively addressed by a single country. However, Iceland and Slovenia‘s domestic-orientated curriculums do address these areas, again highlighting the influence of what a curriculum is called on curriculum content.

Nevertheless, food education interventions are beginning to combine practical-based growing and cooking with knowledge-based nutrition [38]. For example, a systematic review examined the impact of garden-based nutrition programmes on children‘s nutritional outcomes [40], but more research into comprehensive food education is needed.

### 4.2. Food Literacy Findings

#### 4.2.1. Curriculums Predominantly Focus on Confidence and Empowerment with Food (Theme 1) and Significantly Less on Social-Cultural, Equity and Sustainability Themes (Themes 2 and 3)

Analysis of how countries address each FL theme highlights interesting patterns.

Regardless of how comprehensively countries address FL, they all focus on Theme 1, particularly the Nutrition Knowledge subgroup. Furthermore, when a country like Denmark has only a very basic food education, it defaults to Nutrition Knowledge. Further analysis shows all countries except Japan, Ireland, Slovenia, and Czech Republic significantly over-index on Nutrition Knowledge with curriculums overlapping, as typified in England. This could be considered as reinforcing health messaging, but in fact leaves critical FL competencies unaddressed. This approach reflects food education literature with most studies similarly focused on healthy eating and nutrition [88,89,90,91].

Food Preparation Skills, a subgroup of Theme 1 which divides cooking into six competencies, was unexpectedly covered to some degree by all countries, except Ireland. A similar emphasis on cooking was found in the literature, but this finding contrasts the popular notion that children are not taught to cook at school [92]. Each country covered practical food skills to varying degrees, but how well this was addressed was significantly influenced by the policy language and level of curriculum detail. Sweden‘s HCS illustrates this lack of detail, vaguely stating children learn “planning and organising the preparation of meals” [93] (p. 43), Contrastingly, Slovenia‘s Household curriculum addresses every single competency in detail.

This demonstrates how the FL framework can extrapolate detail from curriculums, dispel myths about cooking in education and bring insight to where the issue may lie. For “learning to cook” is included in curriculums, but explanations of what this involves, and classroom implementation may be inadequate, especially given generalist primary teachers lack confidence in practical subjects [16,94].

The socio-cultural competencies in Theme 2 are hardly addressed in most countries with notable exceptions in Australia, Scotland, and Norway. The Enjoying Cultural Foods subgroup is most frequently included, but an overall lack of comprehensive coverage highlights how sociocultural food issues are often overlooked in policymaking [95]. This reinforces FL‘s suitability as a comprehensive food education term, as the only term to explicitly include sociocultural topics.

There is a similar lack of coverage of Equity and Sustainability for Food Systems (Theme 3). Australia, Scotland, and Ireland clearly connect food with the environment and social justice, whilst Japan and England do not address any competency within the theme. The remainder do sometimes address sustainability and corporate influence, but rarely comprehensively. Although the literature identified sustainability as an emerging food education field [44,96], the lack of policy inclusion is unsurprising, as connecting the FS with environmental impacts is a relatively recent but growing practice [57].

That said, there is significant potential to better address Theme 3. Multiple curriculums included sustainability, biodiversity, social justice, economics, politics, and waste; however, their lack of connection to food meant it was not scored. Connecting topics to food and the FS could help address FL curriculums far more thoroughly.

#### 4.2.2. Primary Curriculums Rarely Deal with FL Comprehensively but Have Potential to

Whilst food education is present in curriculums, it is rarely considered as a whole, comprehensive subject. The specialist, siloed nature of food education literature which focuses, for example, on health, cooking, or growing food (see Appendix A, Table A1) also demonstrates a lack of comprehensive approach to food education. Analysis shows no single country or primary curriculum addresses every single FL competency comprehensively, despite positively selecting countries for their mandatory inclusion of food education. However, all FL competencies were addressed somewhere in current curriculums analysed. This shows it is possible to create a food education curriculum that addresses FL comprehensively and teaches primary children to navigate the FS. A comprehensive, curriculum-based, food education is possible in primary school, even if no government achieves this yet.

#### 4.2.3. Food Literacy Is Most Comprehensive When Delivered through a Single Food Curriculum, Complemented by Non-Food Curriculums

Findings show FL is most comprehensive when food is taught through a single food curriculum, rather than split across multiple curriculums, as in Norway and Scotland. Their single curriculums broadly cover all FL themes, connecting food with health and the environment, combining practical skills with knowledge, in a detailed curriculum. Non-food curriculums complement a core food curriculum, addressing supplementary, specialist competencies such as nutritional composition in science, or local FS in geography to make FL as comprehensive as possible and helping children navigate the FS for life. However, this relies on successful policy implementation.

### 4.3. Impact of Policy Approach on FL

Food education was found in mandatory curriculums of all 11 countries for the duration of primary school. Analysis shows when food education is consistently included for the duration of the primary curriculum, with clear learning progression for each year group, it simply gives more opportunities to address more FL competencies, thereby better addressing FL. This contrasts literature analysing food education interventions, finding them to be short, fixed term, and outside of the curriculum [97].

Where food education is located within the wider curriculum, also influences how FL is addressed. When curriculums are designed under health, domestic, design, or vocational perspectives, it affects the range and number of FL competencies included. Food employed as design material in D&T makes it a vehicle for learning about design. Food knowledge is merely a supplementary outcome, much like when food is used to teach mathematics [48]. Instead, FL is best addressed when food knowledge and skills are intended learning outcomes, as in Norway.

How a curriculum is structured significantly affects how FL is addressed. Analysis revealed FL to be most comprehensive when food is taught through a single curriculum, as in Norway, rather than splitting it across curriculums. Scotland and Australia‘s approach using two curriculums also resulted in more comprehensive FL; however from a policy development perspective, careful consideration is needed to ensure the curriculum is free from duplication and addresses a wide range of FL topics. Contrastingly, England’s less considered use of two curriculums for food education results in an over index on health competencies [98] leaving many areas of FL unaddressed, and is an illustration of how spreading food across multiple curriculums can be detrimental to addressing FL.

An alternative policy approach could be to use curriculum structure through implementation of cross-curriculum themes, to support comprehensive FL and ensure FL topics are considered within all subjects. Australia and Norway uniquely use this approach to bring sustainability, health, and cultural topics into all subject curriculums, thereby addressing more Theme 2 and 3 competencies (Joy and Meaning through Food and Equity and Sustainability for Food Systems).

Structure within curriculums including progression of learning and policy detail is an important feature in how FL is addressed. The number of explicit skills and knowledge progression in food curriculums was linked strongly to their FL score. Australia‘s HPE curriculum establishes progression by teaching “always” and “sometimes” foods in Foundation Year, progressing to media health message analysis in Year 6. Whereas England‘s D&T Cooking and Nutrition builds in limited progression across just 2 key stages, which the literature substantiates [16].

Equally, the level of detail within the policy significantly affects how a country addresses FL. As England demonstrates, a lack of curriculum detail influences FL scores, whereas curriculums such as in Slovenia, explicitly break down large subjects such as health and cooking into smaller topics with specific detail about learning outcomes, consequently addressing FL more comprehensively. This requires less teacher interpretation whilst potentially impacting policy implementation by generalist primary teachers who lack food skills and confidence [16,94].

### 4.4. Study Limitations

The high-income countries selected for this study represent a limitation of this research, as this selection excludes examples from low- and middle-income countries and only represents a small sample of international curriculums. Methodologically, managing curriculums across international education systems led to variation in age groups where curriculums applied, and the application of Google Translate may also have lost some language nuance, possibly impacting findings. The impact of only one researcher conducting content analysis means that findings were not corroborated and findings do not consider the impact of other food education approaches outside of curriculum policy. Food education policy implementation, including whether topics are delivered as mandated and the curriculum inspection programmes in each country were not examined. These plus, analysis of food education curriculum impact on pupil’s food knowledge, behaviour and health outcomes could be directions for future research.

## 5. Conclusions

This study demonstrates how 11 different countries approach food education policy and address FL in primary curriculums. FL, employed for its broad and comprehensive coverage of competencies required to eat well for life, recently reflecting FS thinking, was successfully used as a framework to analyse education policies from different countries.

How countries use food education within primary education policy to address FL, varies enormously. All have dedicated food curriculums, supported by non-food curriculums, but there is no consensus as to what food education is called or constitutes. Most curriculums focus on Confidence and Empowerment with Food, with many over-indexing on Nutrition Knowledge. Most countries inadequately address the sociocultural, equity, and sustainability competencies.

No single country or curriculum addresses FL comprehensively but the fact that all competencies were addressed by existing primary curriculums, indicates it is possible. Food literacy is most comprehensively addressed when a single curriculum is dedicated to food learning outcomes, includes food knowledge and skills progression, and is rich in policy detail. Food literacy was most comprehensively addressed in Norway and Scotland through curriculums which addressed all three FL themes, using a combination of practical skills and knowledge, across a comprehensive range of food topics. However, insights can also be gained from every country thanks to their unique approaches, simultaneously indicating a lack of international, standardised food curriculums.

The lack of comprehensive approach from any curriculum in any country indicates that food education is not yet fully leveraged as one of the many policy tools needed to address the global FS challenges of climate change and improving diets [5] that children will inevitably face. This analysis begins to address the significant gap in primary food education policy research.

## Figures and Tables

**Figure 1 ijerph-19-02019-f001:**
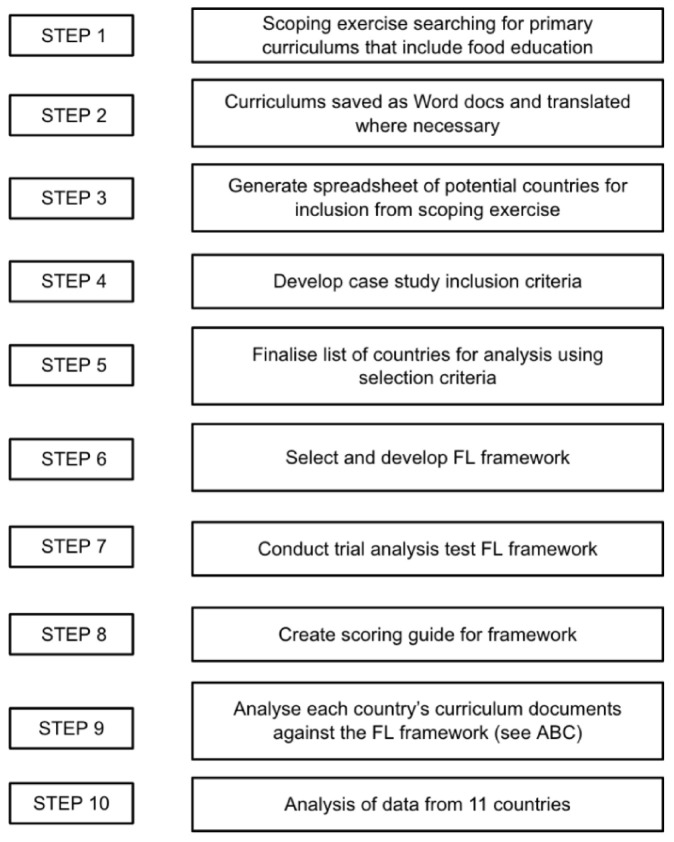
Research steps overview. Source: authors.

**Figure 2 ijerph-19-02019-f002:**
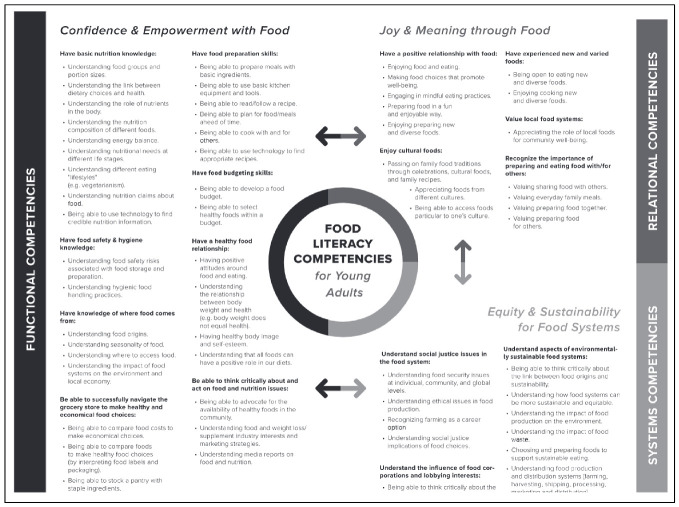
Food Literacy Competencies for young adults grouped by three FL themes [76].

**Figure 3 ijerph-19-02019-f003:**
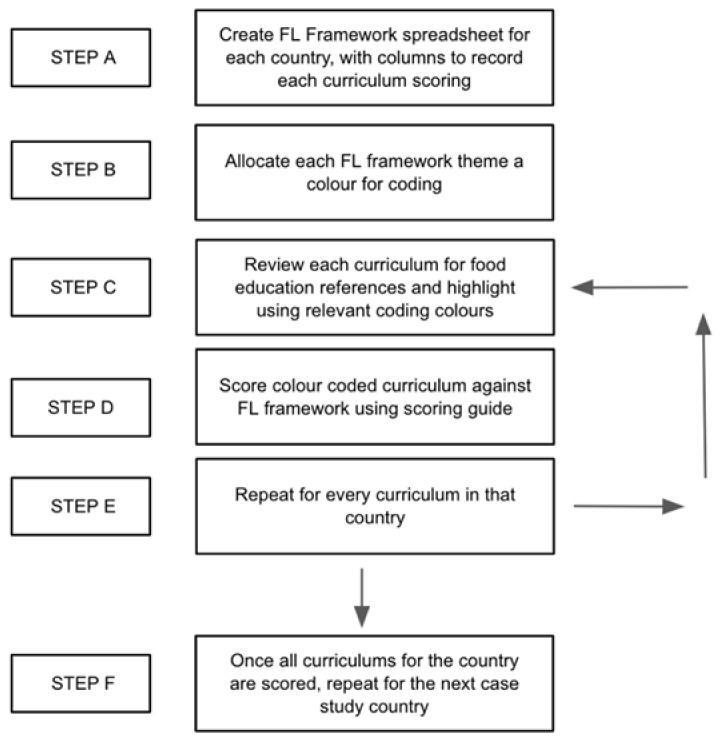
Curriculum content analysis process steps. Source: authors.

**Figure 4 ijerph-19-02019-f004:**
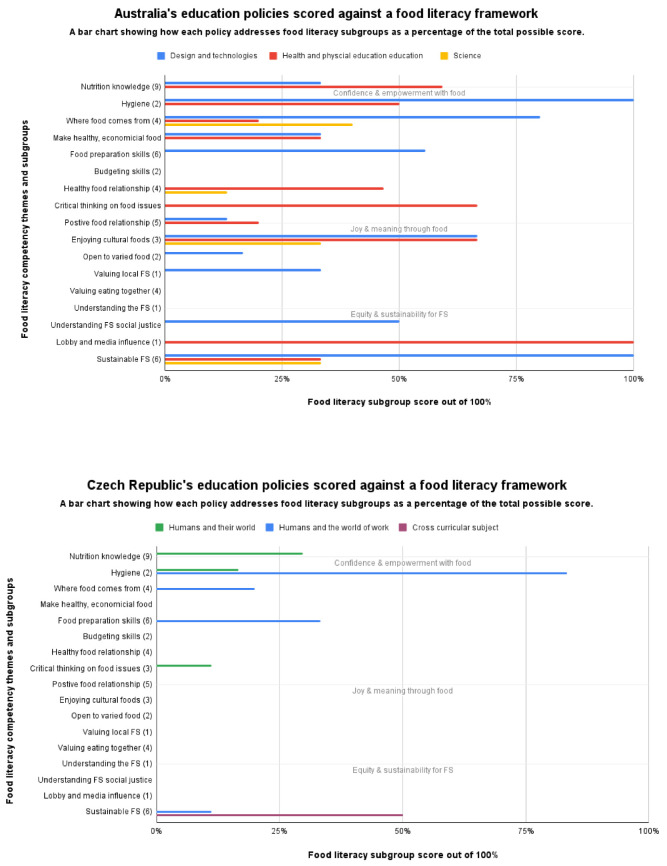
Overview of how 11 countries address FL. Confidence and Empowerment with Food, Joy and Meaning Through Food, and Equity and Sustainability for Food Systems refers to FL framework themes. Source: authors.

**Table 1 ijerph-19-02019-t001:** Food education terms defined. Source: authors.

Food Education Term	Definition
Home Economics	“at school, the study of cooking, sewing, and subjects relating to the management of a home” [18]
	“a curriculum…to discover and further develop their own resources and capabilities to be used in their personal life, by directing their professional life, preparing them for life” [19]
Food Technology	“knowledge and skills to design and make food products effectively…use the physical, chemical and nutritional properties of foods to meet a specific need…implement their design hygienically, safely and effectively. They need to evaluate the design and the product” [20]
Nutrition Education	“any combination of educational strategies accompanied by environmental supports, designed to facilitate voluntary adoption of food choices, and other food and nutrition-related behaviours conducive to health and wellbeing (of individuals, community, planet)” [21]
Food and Cooking Skills	“a wide range of skills required to feed families, including not only factors involved with the meal preparation… but also knowledge of how to plan and budget for food and organise and plan meals that other members of the household will find acceptable” [22]
Food Education	“Education that supports learning about food, nutrition and the role that food plays in one’s life, relationships, culture, communities, environment, and in history and society” [23]
Food Literacy	“the ability of an individual to understand food in a way that they develop a positive relationship with it, including food skills and practices across the lifespan in order to navigate, engage, and participate within a complex food system. It’s the ability to make decisions to support the achievement of personal health and a sustainable FS considering environmental, social, economic, cultural, and political components” [24]

**Table 2 ijerph-19-02019-t002:** Inclusion and exclusion criteria. Source: authors.

To Be Included within the Case Study, Curriculums Must:	Justification
Have an English translation.	This is the lead author’s only language.
Be issued by national government departments/applied in national education systems.	Ensures a consistency across levels of governance, improving rigor and reliability.
Be implemented in education for 4–11-year old’s.	Research scope is primarily concerned with primary education, which determines the age range.
Be mandatory and currently implemented curriculum.	To make recommendations to policymakers, policies need to be from national governments where live policy implementation is mandated.
Include any kind of “food education”.	To establish where food education takes place, curriculums must refer to food topics.
**Excluded scenarios**	**Justification**
Nursery, pre-school, secondary and tertiary education.	Research is only concerned with primary education.
Policy implementation and effectiveness.	This makes the research scope too extensive.
School meals.	Not typically included within education policy and keeps research scope manageable.

**Table 3 ijerph-19-02019-t003:** Scoring guide. Source: authors.

How Well Does the Curriculum Address the FL Competency?	Explanation	Score
Comprehensively	Fully addresses the FL competency with detail.	3
Partially	Addresses the FL competency, but only in some way, lacking detail.	2
Acknowledges competency	Makes references to the competency but is vague and lacking in detail.	1
Fails	Makes no reference to the FL competency.	0

**Table 4 ijerph-19-02019-t004:** Overview of curriculums in each country and province that include food education and what they are called. Grouped into food curriculums, either practical or health approaches and non-food curriculums. Source: authors.

Location (Number of Curriculums That Met Inclusion Criteria)	Food Curriculum		Non-Food Curriculum		
	Practical Approach	Health Approach	Science	Other	
**Australia (3)**	D&T	Health and Physical Education	Science		
**Czech Republic (3)**	Humans and the World of Work		Humans and their World	Cross Curricular Subjects	
**Denmark (2)**		Health and Sexual Education and Family Knowledge	Nature/ Technology		
**England (3)**	D&T	Relationships and Sex Education and Health Education	Science		
**Iceland (3)**	Home Economics		Natural Sciences	Social Science	
**Ireland (3)**		Social, Personal and Health Education	Science	Geography	
**Japan (4)**	Home Economics		Science	Living Environment Studies	PE
**Norway (2)**	Curriculum in Food and Health		Science		
**Scotland (4)**	Technologies	Health and Well Being	Science	Social Studies	
**Slovenia (2)**	Household		Knowledge of the Environment		
**Sweden (3)**	Home and Consumer Studies		Science	Geography	

**Table 5 ijerph-19-02019-t005:** Food education topics covered in science. Tick indicates country includes the topic within science. Source authors.

Science Topics	Australia	Czech Republic	Denmark	England	Iceland	Ireland	Japan	Norway	Scotland	Slovenia	Sweden
Health		✓	✓	✓	✓	✓		✓	✓	✓	✓
Diet		✓		✓		✓		✓	✓	✓	✓
Senses	✓		✓	✓		✓		✓	✓		✓
Plants	✓	✓	✓	✓	✓	✓	✓	✓	✓	✓	✓
Animals (including humans)				✓			✓				✓
Hygiene		✓	✓	✓	✓				✓	✓	
Nutrition		✓		✓	✓						✓
Local foods					✓						
Food science	✓										
Sustainability	✓							✓			

## Data Availability

The data used in this study is available upon request.

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
