# Peer review of "How Primary School Curriculums in 11 Countries around the World Deliver Food Education and Address Food Literacy: A Policy Analysis"

_ijerph, 2022, doi:10.3390/ijerph19042019_

Round 1

Reviewer 1 Report

Language and technical care:

The manuscript requires extensive attention in terms of the following language and technical aspects:

  • Page 1, line 27 – Should the word ‘International’ be used as a stand-alone keyword?
  • Page 1, line 9 and line 34 – Can the Food System ‘deliver’ affordable, healthy, sustainable diets, or does it play an important role in influencing such diets.
  • Page 2, line 53, 56 and 57 – Check consistent use of caps/non-caps in ‘Food Education’.
  • Page 2, line 65 – title of Table 1 states that it is ‘created by author’, yet more than one author is listed at the beginning of the article.
  • Page 2, table 1 – Check the use of full stops in the definitions for consistency.
  • Page 2, table 1, definition of Nutrition Education – Close bracket.
  • Page 2, table 1, definition of Food Education – ‘one’s life’ and not ‘one”s life’;
  • Page 2, line 66 – To improve readability, insert a space between the end of the table and the next paragraph starting at line 66.
  • Page 3, line72, also page 4, lines 122 and 124 – the correct spelling is ‘Maths’ and not ‘Math’s’.
  • Page 3, line 84 – Check the spacing between ‘Literacy’ and ‘literature’.
  • Page 3, line 93 and 94 – Check spacing between reference numbers for consistency.
  • Page 3, line 96 – Full stop missing at end of sentence.
  • Page 3, lines 100, 107 and 115 – To aid in readability, consider changing the word order of ‘cooking, growing, gardening and health’ to ‘gardening and growing of fruit and vegetables, cooking, and health’. As it is, the word ‘growing’ appears without context.
  • Page 4, line 139, 149, 151 – Check spacing between reference numbers for consistency.
  • Page 4, line 147 – The sentence ends without any punctuation mark.
  • Page 6, Figure 1, Step 4 – ‘word docs’ should be ‘Word docs’.
  • Page 6, Figure 1, Step 12 – 13 countries are mentioned while in all other references the number of countries n=12.
  • Page 6, line 212 and 213 – Use commas between a list of words.
  • Page 7, line 220, also Page 8, line 246 – There seems to be a problem with the Table numbers; since there is no table 2, all other tables and Table Headings are incorrect.
  • Page 7, Table – Check caps at the start of sentences for consistency.
  • Page 7, Table – Under ‘Justification’ consider adding the word ‘education’ after primary.
  • Page 7, line 237 – Punctuation at the end of the sentence is missing.
  • Page 8, line 242 – Are there two references by Slater et al. for Food Literacy competencies, and are they both mentioned in the one reference, OR do the authors actually mean ‘Slater et al.’s FL competencies in Figure 2 as well as additional competencies were converted into a FL framework…’
  • Page 8, Figure 2 – The font size and colour make this figure extremely difficult to read. Since there is a lot of important information in this figure, it should be easy to read.
  • Page 18, line 371 – Please check whether it is ‘30 out of 62’, or ‘30 of 62’.
  • Page 18, line 388 – Remove the additional space between ‘topics.’ and ‘Australia’s…’
  • Page 19, line 435 – Consider ‘it was unexpected to discover all countries under investigation…’.
  • Page 20, line 473 – Remove the full stop after ‘Technologies’.
  • Page 21, line 511 – The sentence starting with the word Regardless is not the first sentence in the paragraph and should not be indented.
  • Page 22, line 557 – Remove the additional space between ‘FS.’ and ‘A’.
  • Page 22, line 577 – Add the word ‘it’ before the word ‘also’.
  • Page 23, line 610 – Change ‘country’s’ to ‘countries’.
  • Page 23, line 624 and 625 – Remove additional spaces before ‘Food Literacy’ and between ‘through’ and ‘curriculums’.

The manuscript is very well referenced using relevant, up-to-date references.

Literature Review:

This reviewer identified no shortcomings in the literature review – it is up to date and comprehensive.

Methodology and materials:

The reviewer believes that the methodology is fairly straight forward and explained well, and that good and acceptable research procedures have been followed. Since the paper is already fairly lengthy, it is understandable why detail has been limited in some areas. However, a little more detail regarding the content analysis of the curriculums and what methods were used would have been helpful.  

On Page 7, the authors state that in order to be included curriculums should have had an English translation, but they also mention that Google Translate was used to translate some of them. This should be clarified for the sake of consistency.

Results and Discussion:

The reviewer believes that the study results are of high quality and value, particularly in context of the growing nature of eating behaviour and consequent health as a result of food choices among primary school children.

Conclusion:

In terms of the study limitations on page 10, line 271 (see also page 19, line 427) the reviewer believes the authors should be more specific about the 12 countries they chose for their sample, since there may be a limitation in terms of developed versus under-developed countries.

The reviewer believes that the conclusion is well written and the worthiness of the research is evident, particularly in light of our never-ending fight against the global onslaught of diet-related illnesses.

Author Response

Thanks for your comments. Please see attachment. 

Reviewer 2 Report

The focus on food literacy was interesting, but it was often hard to follow what the authors did and found. It may be helpful if the authors consider improving the structure and the flow of information as the manuscript seems disjointed. Also, there are spelling and grammatical issues as well as incorrect (or missing) punctuations that should be addressed.

Abstract

The authors should consider putting FL into parenthesis after the phrase “food literacy” before they use the abbreviation in the rest of the text. Also, is it appropriate to say curriculums (instead of curricula) in academic writing/journal articles? The authors may want to double-check the best phrase to use in the abstract, the title, and throughout the manuscript or follow the journal’s preference.  

Introduction

  • Page 1, line 39: What does “stronger measures” refer to? Measures of what? The authors should provide some clarity.
  • Page 1: The authors have identified a number of policy levers, so can they provide more justification for the focus on education?
  • I am concerned that the authors seem to oversimplify the links between schools, childhood, and food education. There are many children who are not in school or who attend school in very poor conditions. Also, there may be few teachers with the adequate knowledge/training to engage in delivering food education, especially when they are swamped with teaching other materials. Additionally, education alone (with a focus on children) without facilitating easy and affordable access to food and considering other factors that affect food choice and food systems may be futile. At what point do the authors acknowledge that these challenges exist globally?
  • Can the authors explicitly state the link they see between food literacy and the transformation of the food system?
  • For Appendix 1, can the authors indicate the countries or continents where the papers were published/where the research was done? Also, how did they narrow down the applicable papers in the first 50 results? Additional details in the table may provide some context for the reader.

Materials and methods

  • Can the authors provide more details about the scoping method? Also, can they describe the selection criterion that is mentioned on page 5?
  • What do the authors mean when they say, “the researcher’s school experience”? (pages 6-7).
  • The research steps are confusing. Initially, I thought the authors found certain curricula online, which is how they were able to narrow down to 12 countries in step 1. However, it seems the curricula were obtained after the countries were identified (in step 2). Can the authors clarify the process? Since there are actual curricula that are used in various classrooms, it is unclear why the authors chose to refer to education-related policy documents as curricula.
  • The following sentence sounds awkward: “A database organised potential countries and their curriculums (step 3).” Perhaps, the authors could say, “we used a database to organize potential countries…” Also, did the authors use a database or a spreadsheet, as indicated in Figure 1 (step 3)?
  • Testing study design, framework, etc. doesn’t seem to be enough to use the term “pilot study”. Can the authors describe what was actually done?
  • The authors mentioned 13 countries in Figure 1 but said they had 12 countries within the text. Can they confirm which is the correct number?
  • How many countries/states were originally on the list before the authors narrowed the number down to 12? Also, all the current locations appear to be high-incomes countries/states. Does this mean the authors could not identify any low/middle-income countries that had relevant curricula available?
  • The following sentence needs either some punctuation or revision: “The main limitation in this research is how only one researcher carried out content analysis meaning findings were not corroborated.”
  • I don’t understand why the authors chose to do the following: “the number of competencies in the group was multiplied by the maximum scoring criteria score of 3 and a percentage was calculated to allow for variation in the number of competencies in each subgroup.” Can the authors explain the rationale behind this approach?
  • It is strange to see study limitations so early in the manuscript. The authors should consider moving this section towards the end.

Results

  • For table 5 and Appendix 3, it will be helpful if the authors spell out the various abbreviations that they used.
  • I don’t quite follow how the authors arrived at the scores provided in Table 5. What is the difference between the overall and comprehensive scores?
  • I struggled to follow the graphs for each country because I had to keep going back and forth to see if I could identify similarities/differences across countries. Have the authors considered finding other ways to present the data so that the reader can see the score across nutrition knowledge for all countries at once, for instance?
  • I was surprised to see a hypothesis mentioned on page17 when it had not appeared earlier in the manuscript. Unless I missed it, can the authors explain why they did not propose the hypothesis in the introduction or methods sections?
  • For table 5, the authors make reference to the color green, but that color doesn’t exist in the table.
  • Cultural preferences for food may affect the types of information that children receive in school. Have th

Discussion

In general, the authors should be careful not to make sweeping statements about food literacy being comprehensive, etc. because they had access to some curricula and not all of them. It possible that the countries they focused on may have other approaches through which children learn about food and food literacy, which may not be captured in the documents examined through this study. Additionally, each country may have a unique approach to food education because of differences in food culture, how food is perceived, etc. which can even differ widely within one country. Have the authors considered how this may be linked to their findings? What I am referring to here is different from the socio-cultural competencies mentioned in the appendix.

Reviewer 3 Report

The article studied food education within primary school education policy in 12 countries (England, Australia, Denmark, Norway, Scotland, Iceland, Japan, Ireland, Ontario, Slovenia, Sweden, and Czech). They used case study methodology with a Food Literacy framework to analyze the curriculum policy contents. The authors concluded that the food education policy to address food literacy (FL) varies enormously among the 12 countries, which all have dedicated food curriculums supported by non-food curriculums. They suggested “The most comprehensive are single, 24 detailed food curriculums, complemented by non-food curriculums where food knowledge and 25 skills progress clearly and are the intended learning outcome.”

This is a well written article; however, why did the 12 countries selected? The authors may give an introduction.

Why are some representative countries not included in the analysis? For example, India, China, the United States, …

Author Response

(The authors gave the same response as above.)

Reviewer 4 Report

I appreciated the goal of the paper: comparing 12 countries, the Authors intended to evaluate the role achieved by the schools' policy, through "food education", in creating a complete "literacy" of every pupil about his/her relationship with the food in the most complete and global way. It is not only nutrition, health, but also technology, cooking capacity, pleasure, gastronomy, conviviality, etc. The results for the Authors, after examining the situation of the different countries, are quite disappointing because food education is in general partial, incomplete and not reaching the proposed goals (overweight and obesity are still too frequent, food literacy of people quite insufficient), but the message of the paper to policymakers is very strong: reaching a complete food literacy is necessary for a complete and mature life, physically and mentally, therefore the school must work for the achievement of a complete food literacy.

My only criticism is about the choice of the countries to be examined: all the scandinvian countries, all parts of UK, some far away countries as Japan and Australia, Ontario as the only representative for the North American continent, only Slovenia for continental Europe: States full of people and of long educational and gastronomic tradition as China, South America, Germany, France, Spain and Italy, all the Middle East, all Africa -  excluded. Why?

Author Response

(The authors gave the same response as above.)
